# Crosstalk between Interleukin-1β and Type I Interferons Signaling in Autoinflammatory Diseases

**DOI:** 10.3390/cells10051134

**Published:** 2021-05-08

**Authors:** Philippe Georgel

**Affiliations:** Laboratoire d’ImmunoRhumatologie Moléculaire, Institut National de la Santé et de la Recherche Médicale (INSERM) UMR_S 1109, Institut Thématique Interdisciplinaire (ITI) de Médecine de Précision de Strasbourg, Transplantex NG, Faculté de Médecine, Fédération Hospitalo-Universitaire OMICARE, Fédération de Médecine Translationnelle de Strasbourg (FMTS), Université de Strasbourg, 67085 Strasbourg, France; pgeorgel@unistra.fr

**Keywords:** inflammation, type I interferons, interleukin-1β, crosstalk

## Abstract

Interleukin-1β (IL-1β) and type I interferons (IFNs) are major cytokines involved in autoinflammatory/autoimmune diseases. Separately, the overproduction of each of these cytokines is well described and constitutes the hallmark of inflammasomopathies and interferonopathies, respectively. While their interaction and the crosstalk between their downstream signaling pathways has been mostly investigated in the frame of infectious diseases, little information on their interconnection is still available in the context of autoinflammation promoted by sterile triggers. In this review, we will examine the respective roles of IL-1β and type I IFNs in autoinflammatory/rheumatic diseases and analyze their potential connections in the pathophysiology of some of these diseases, which could reveal novel therapeutic opportunities.

## 1. Introduction

Numerous reports have documented the roles of IL-1β and type I interferons (IFNs) in the defense mechanisms that are engaged upon bacterial (such as *M. tuberculosis* [1]) and viral [2] infections. Type I (and type III) IFNs exert powerful antiviral activities that have been extensively described [3,4], while those mediated by IL-1β are more scarcely defined [5]. Furthermore, the interplay of these cytokines and their downstream signaling pathways has also been largely explored during infectious diseases [6], COVID-19 being the most recent example [7].

These cytokines are produced following the activation of dedicated pattern-recognition receptors (PRRs) [8] in response to specific pathogens and the associated molecular patterns (PAMPs) that they express. Interestingly, the same PRRs (nucleotide-binding oligomerization domain-like receptors—NLRs, Toll-like receptors—TLR or AIM2-like receptors—ALRs) are also activated upon the detection of danger signals (DAMPs [9,10,11]) produced in sterile conditions. In this case, inflammation, instead of creating the appropriate conditions to clear off an invading pathogen, generates tissue damage and evolves towards detrimental endpoints for the host. First, this review will provide some examples of autoimmune/autoinflammatory diseases that are caused by the deregulated expression of type I IFNs and IL-1β. Indeed, these cytokines are major mediators of inflammation and can be incriminated in many cytokinopathies [12], which are diseases caused by alterations in a single gene affecting cytokines expression. Several examples of interferonopathies and inflammasomopathies will illustrate these cases. Additionally, type I IFNs and IL-1β perturbations can also result from interactions between many genes and the host environment. Lupus, a disease in which patients exhibit an “IFN signature” [13] (i.e., overexpression of a subset of IFN-stimulated genes) and Alzheimer’s disease, during which IL-1β is known to be overexpressed [14], will serve as examples for such complex (multigenic/multifactorial) diseases in which these cytokines are involved. Next, we will analyze several cases where reciprocal interactions between them have been observed, and the therapeutic perspectives that have been derived from these observations. Multiple sclerosis, a disease treated with IFN-β (among other therapeutic options) and which is also characterized by increased IL-1β expression, will be described. In parallel, gout and rheumatoid arthritis (RA) are joint inflammatory diseases in which reducing IL-1β overexpression can represent an efficient therapeutic opportunity. Interestingly, promoting type I IFNs expression recently appeared as an attractive way to dampen IL-1β production in animal models for gout and RA [15,16]. These examples in which type I IFNs and IL-1β exert a reciprocal control will reveal novel options to treat patients suffering from these inflammatory diseases, whose general features are given in Table 1. Finally, innovative cell culture methods designed to investigate and aimed at deciphering these interactions between cytokines at the molecular and cellular levels will be discussed in a prospective chapter.

## 2. Type I IFNs in Autoinflammation

Since their initial discovery in 1957 [17], type I IFNs have been essentially considered beneficial with regards to their unique antiviral activities [18]. More recently, however, it appeared that the deregulated and inappropriate expression of these cytokines could be harmful. Indeed, in the absence of any obvious viral trigger, the overexpression of type I IFNs was noted in patients suffering from inflammatory disorders [19], some of which were caused by single-gene mutations (monogenic diseases), while others are classified within complex diseases, i.e., requiring environmental factors and many specific genetic alterations to promote pathogenic features. *STING*-associated vasculopathy with onset in infancy (SAVI) belongs to the first category of ailments and is caused by a gain-of-function mutation in the *STING* gene; this gene encodes a protein that is at the cross-roads between the cGAS (cyclic GMP-AMP synthase, an exogenous DNA sensor) and the interferon regulatory factors (IRFs)-3 and -7, which induce type I IFN transcription [20]. In these patients, TANK-binding kinase (TBK1) is constitutively activated in the absence of viral RNA, leading to spontaneous and massive type I IFN production. Fortunately, Janus kinase inhibitors might be promising drugs to block the signaling pathway downstream of the type I IFNs receptor (IFNAR) and provide relief to a subset of patients with SAVI syndrome [21]. In past years, many additional genetic origins of type I interferonopathies were elucidated following whole-exome sequencing in patients and controls in families affected by these rare symptoms [22,23].

On the other hand, systemic lupus erythematosus (SLE), with the exception of childhood-onset SLE, is a complex disease driven by a combination of genetic, epigenetic and environmental factors [24]. Of note, a hallmark of SLE is the so-called “IFN signature”, describing the overexpression of IFN-stimulated genes (ISGs) in circulating mononuclear blood cells or target tissues [25]. Of note, the level of ISGs expression appears correlated with disease severity [26]. Interestingly, ISGs overexpression is also observed in other inflammatory diseases, such as rheumatoid arthritis [13]. More recently, single-cell RNAseq technology enabled a precise description of gene expression in SLE patients that appeared to form a more heterogeneous population than previously suspected [27]. Such stratification of patients with multiOMICs technologies already sets the grounds for more targeted, individualized therapies [28]. In the frame of the present review, it is noteworthy to observe that, in addition to type I IFNs, IL-1 family member expression can also be used as a biomarker in SLE patients [29]. Finally, severe complications occurring in lupus patients, such as macrophage activation syndrome or pericarditis, have been successfully reduced with anakinra, an IL-1b antagonist [30,31], showing that both IFNs-I and IL-1 participate in lupus pathogenesis, at least in a subset of patients.

## 3. IL-1β in Autoinflammation

Similar to type I IFN-dependent diseases, many inflammatory syndromes result from uncontrolled IL-1β expression. Among them, inflammasomopathies are a group of monogenic diseases caused by hereditary defects in inflammasomes components. Inflammasomes are intracellular multiprotein complexes composed of a sensor (detecting pathogen-associated molecules, such as peptidoglycans from Gram-positive bacteria or sterile components, such as silicate or urate crystals), an adaptor (ASC for apoptosis-associated speck-like protein containing a caspase recruitment domain) and the Caspase 1. Following multimerization of this complex, activated Caspase 1 cleaves pro IL-1β into its mature, bioactive form, which is exported out of the cell through pores formed by GasderminD [32]. The prototypical inflammasomopathy with periodic fever is familial mediterranean fever (FMF), a disease caused by mutations in the *MEFV* (mediterranean fever) gene encoding the protein PYRIN, which is part of the inflammasome complex. Gain-of-function mutations in the *MEFV* gene lead to increased Caspase 1 activation and IL-1β levels [33]. The development of IL-1β antagonists has considerably improved the management of these patients [34].

In addition to these monogenic inflammatory diseases, emerging evidence suggests that IL-1β is also involved in complex neurological disorders, such as Alzheimer’s disease (AD) [35]. Indeed, AD occurrence depends on many factors, such as age, comorbidities, genetics and education level. However, a strong correlation between AD and reactive oxygen species (ROS) production has been evidenced, where ROS are major inducers of NLRP3-dependent IL-1β production [36], including in neurons [37]. Importantly, this observation has led to novel therapeutic options for neurodegenerative disorders affecting an increasing number of patients worldwide.

## 4. Interplay between Type I IFNs and IL-1β in Inflammatory/Autoimmune Diseases

Whilst interferonopathies and inflammasomopathies may appear as very divergent or even antagonistic inflammatory diseases (although an overlap can be observed in some instances, as mentioned in the previous chapters), the pathogenesis of some inflammatory conditions clearly involves both type I IFNs and IL-1β. Multiple sclerosis (MS) belongs to this category, since IL-1β is strongly implicated in this inflammatory, neurodegenerative disease [38], and IFN-β is still a classical first-line therapy [39], although rituximab (an anti-CD20 monoclonal antibody designed to induce B cell ablation) was shown recently as a promising option [40]. Low STING-dependent type I IFNs expression in peripheral blood mononuclear cells (PBMC) isolated from MS patients [41] is in agreement with these observations.

The mechanism by which IFN-β exerts its anti-inflammatory actions has been partially elucidated [42]. It is now very clear that type I IFNs promote IFNAR-dependent *IL-1Ra* (encoding an antagonist of the IL-1β receptor) and *IL-10* gene expressions. Furthermore, type I IFNs and IL-10 were recently shown to negatively regulate the activation of the NLRP3 inflammasome in a STAT3-dependent manner [43,44,45]. These data support the notion that IL-1β and type I IFNs exert antagonistic activities that have been experimentally tested in various inflammatory settings (collagen-induced arthritis, allotransplant rejection), whereby the beneficial administration of type I IFNs has been documented.

Reduced expression of *NLRP3* was also shown to participate in the anti-inflammatory benefits of type I IFNs in MS [46,47]. This observation also likely accounts for the spectacular therapeutic potential of imiquimod, a TLR7 agonist and strong inducer of type I IFNs, which we observed in a mouse model of acute uratic inflammation [15]. Importantly, our work using this mouse model of gout as well as RA models [16] enabled us to develop a framework in which complex cellular interactions are required to account for the counter-regulatory effects mediated by type I IFNs on IL-1β [48]. Future work using elaborate cell culture systems will be necessary to decipher this cellular dialog, as discussed below. Surprisingly, the regulatory roles of IL-1β on type I IFNs and ISGs expression are more scarcely documented [49], and these experimental cell culture experiments would also be useful to explore this issue. In this regard, the recent observation that IL-1β promotes type I IFN and ISGs expression in bone marrow-derived dendritic cells (BMDC) appears of particular interest [6]. A schematic network of type I IFNs and IL-1β interactions is depicted in Figure 1.

## 5. Therapeutic Consequences

As mentioned above, some overlap may exist between IL-1β and type I IFNs in various inflammatory settings, opening novel therapeutic opportunities.

### 5.1. Targeting Type I IFNs in Il-1β-Dependent Diseases

Type I IFNs-based therapies were developed long ago and were particularly useful in hepatitis C virus-infected patients, despite considerable side effects [50]. In this regard, our strategy to perform epicutanieous application of a cream containing imiquimod, a powerful promoter of IFN synthesis to treat inflamed joints of RA or gout mice, appeared as a promising approach to avoid adverse reactions [15,16]. Importantly, we observed a drastic reduction in neutrophils in the cellular infiltrate following imiquimod application, which also certainly participates in the reduced local inflammation through the limitation of ROS production. Topical imiquimod has been used for 20 years in humans to treat genital warts and skin carcinoma [51]; its pharmacokinetics and precautions for use are well known. Therefore, we believe that our pre-clinical studies advocate for using this drug to treat joint inflammation in RA or gout patients, as well as localized skin inflammation, showing evidence of a massive neutrophilic infiltrate (neutrophilic dermatoses). On the other hand, strategies presently in use or under development aim at reducing the IFN-dependent signaling pathway, for instance, in SLE patients with anifrolumab, a monoclonal antibody targeting the type I IFN receptor subunit 1 [52]. Other tools to reduce IFN signaling are the Janus kinase (JAK) inhibitors (jakinhibs), a novel family of compounds effective in myeloproliferative or autoimmune (such as RA) diseases [53]. Given their antagonism, a rise in IL-1β can be expected in patients with reduced type I IFN production as a result of treatment with anifrolumab or jakinhibs, which might require specific attention, and possibly the need for additional anti-IL-1β therapy.

### 5.2. Targeting IL-1β in Interferonopathies

Jakinhibs are the most promising therapeutic opportunities for patients afflicted by type I interferonopathies [54]. As mentioned above, following IL-1β expression levels might be critical in these critically ill patients.

In addition, IL-1β inhibition might also represent a useful strategy in various inflammatory diseases, including interferonopathies. Indeed, this cytokine is also expressed in the central nervous system, where it mediates pain [55]. Supporting this notion is the observation that psoriatic arthritis (PsA) patients treated with anti-TNF antibodies still experience pain, while joint inflammation is concomitantly reduced [56]. In these patients, and possibly in others treated with TNF inhibitors or more generally experiencing pain as a result of inflammatory reactions, there might be room for IL-1β blockers (canakinumab, anakinra). Finally, it is interesting to note that experiments in the experimental autoimmune encephalomyelitis (EAE) mouse model support the therapeutic potential of IL-1 blockade in MS [57], an approach that has been tested in a very limited number of patients suffering colchicine-resistant familial mediterranean fever (FMF, an inflammasomopathy) and MS [58]. Strikingly, MS symptoms were markedly reduced in these patients.

Altogether, these observations indicate that the management of patients suffering inflammatory symptoms might require a combination of drugs targeting various players involved in the pathogenesis of these diseases. Future work aiming at a better characterization of the interplay between these players is needed to provide more efficient and targeted therapeutic approaches. Some insights aiming at this goal are suggested in the perspectives and conclusions of the present review.

## 6. Perspectives

Although several molecular interactions between type I IFNs and IL-1β have been described (transcriptional induction of *IL-1Ra* and *IL-10* genes upon IFN-β treatment [42]; and reciprocally, increased transcription of *IFN-β* and *ISGs* following IL-1β addition in the culture medium of BMDCs [6]), most studies have been performed in cell cultures where one cell type only has been investigated (dendritic cell, monocytes/macrophages, etc.). This constitutes a fundamental weakness, since these cytokines are produced by different cell types (neutrophils [59], eosinophils [60]) interacting in a specific microenvironment. To gain access to more physiological interactions at the cellular and molecular levels, co-cultures, either in two-dimension systems (Boyden chambers) or even using more complex organoids, need to be developed [61]. As seen in Figure 2, considering only type I IFNs (IFN-α/β) and IL-1β and the five main immune cell types that are able to produce them, upon the TLR7-dependent stimulation (with imiquimod, IMQ) of pDCs, an already complex network of interactions is created, in which the reciprocal effects of these cytokines are presently totally unknown and certainly quite different from what can be observed in monotypic cell cultures. Producing mixed cultures in Boyden chambers (which has been previously performed [62]) could be a good starting point, in which each cell type (for instance, pDC and macrophages, or pDC and neutrophils) could be investigated separately after cell sorting with high-throughput technologies (RNAseq) and analyzed morphologically (apoptosis, NETosis, polarization) in various conditions driving cytokine (type I IFNs or IL-1β) synthesis. Such approaches might be instrumental to better characterize, at the cellular level, the recently described interaction between IL-1β-dependent mitochondrial DNA release and cGAS/STING-dependent type I IFNs secretion [63]. In the future, spheroids or organoids might add complexity to the system by adding support cells such as keratinocytes or fibrocytes and extracellular matrix components.

## 7. Conclusions

We have merely touched on the complexity of inflammation here by analyzing the reciprocal interactions of two cytokines. Despite the paramount importance of IL-1β and type I IFNs in autoinflammatory diseases, many other cytokines, among which TNF are self-evident, can certainly not be neglected. In 2006, Banchereau and Pascual published a seminal paper in which they extended the Th1/Th2 concept into a “compass of immunity and immunopathology” organized into two perpendicular axes: one defined by the reciprocal interactions of IFN-α and TNF and the other by IL-4 and IFN-γ [64]. According to this model, SLE was identified by an overexpression of IFN-α. Fifteen years later, high-throughput technologies have evidenced the heterogeneity of patients suffering from complex diseases such as SLE or RA, which are now defined as pathotypes [65]. Because some cytokines are direct drivers of immunopathology and because the quantification of most cytokines is easily feasible with multiplex technology, we suggest that providing an extensive profiling of cytokines (in the blood or the affected tissue if accessible), a “cytokinome” as suggested by others [66], would be a useful tool to better define patients sub-groups by comparison with a reference of healthy subjects [67]. This approach, illustrated by the “multidimensional compass” illustrated in Figure 3, would also be instrumental in defining the best therapeutic option for a patient, following its impact on the normalization of its cytokine profile and eventually adjusting it. In this example, two RA patients are identified by increased TNF expression compared to a control group (with the reference cytokinome resulting from a set of healthy donors) with variables (age, sex, etc.) matching the patients. However, following anti-TNF therapy, each exhibited a different outcome. In patient 1, the normalization of TNF levels was accompanied by increased IFN-α/β secretion and paradoxical psoriasis (a recently described possible consequence of anti-TNF antibodies [68]), which might require appropriate management (jakinhibs, eventually). On the other hand, patient 2, in which the same treatment also enabled a marked reduction in the circulating TNF level and improvement of joint inflammation, responded by an additional strong decrease in IL-1β expression (as previously described [69]), putting him at risk of developing various microbial infections and therefore requiring specific monitoring in the future. These hypothetical cases indicate that the determination of the cytokinome and its evolution upon treatment might bring substantial benefits to patients with inflammatory diseases.

## Figures and Tables

**Figure 1 cells-10-01134-f001:**
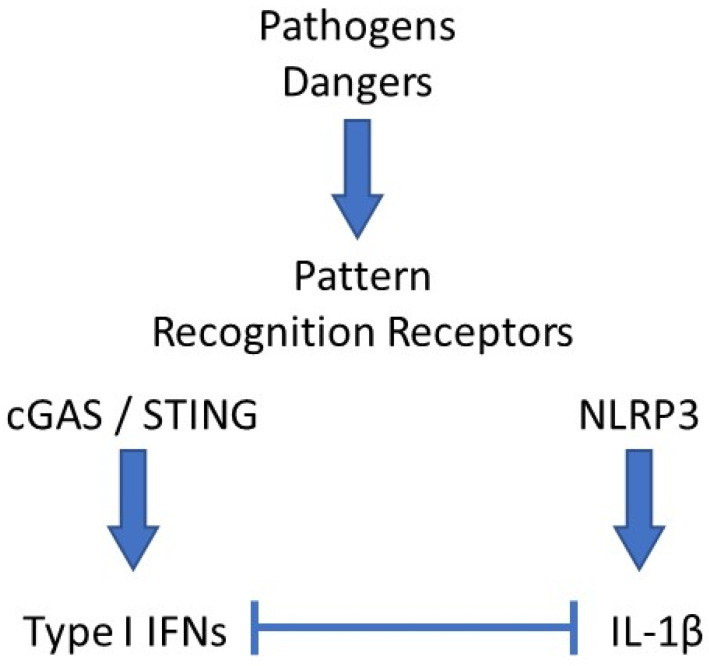
Schematic network of interactions between type I IFNs and IL-1β. Pathogens- or danger-associated molecular patterns (PAMPs, DAMPs) interact with their cognate pattern recognition receptor (PRRs). In the example shown here, DNA binding and activation of the cGAS/STING pathway leads to type I interferons (IFNs) secretion, while monosodium urate (MSU) crystals activate the NLRP3 inflammasome, which induces IL-1β release. In most cases, both cytokines exert antagonistic activities, mutually repressing their expression levels by various mechanisms.

**Figure 2 cells-10-01134-f002:**
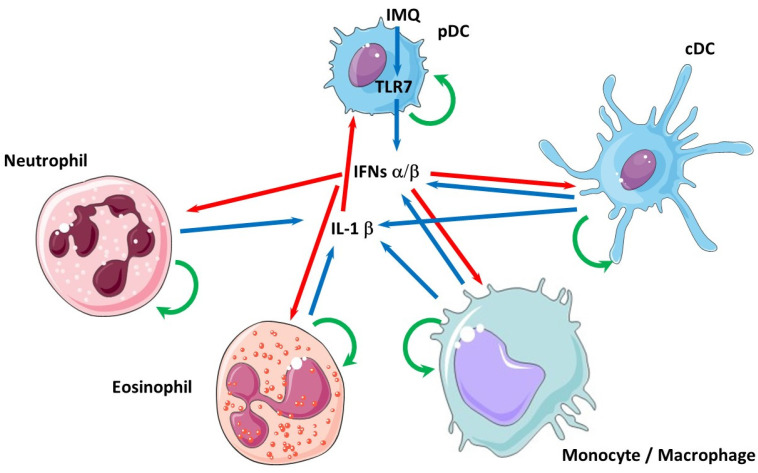
The complex network of interactions between type I IFNs, IL-1β and the cells that produce them. Simplified representation of the potential interactions between plasmacytoid (pDC) or conventional (cDC) dendritic cell, macrophages/monocytes, neutrophils and eosinophils upon, for example, imiquimod (IMQ) stimulation acting via TLR7 in pDCs. Blue arrows denote cytokine expression, red arrows indicate that these cytokines exert an effect (activation or inhibition) on target cells and green arrows represent retro-control of the cytokines on the cells that produce them.

**Figure 3 cells-10-01134-f003:**
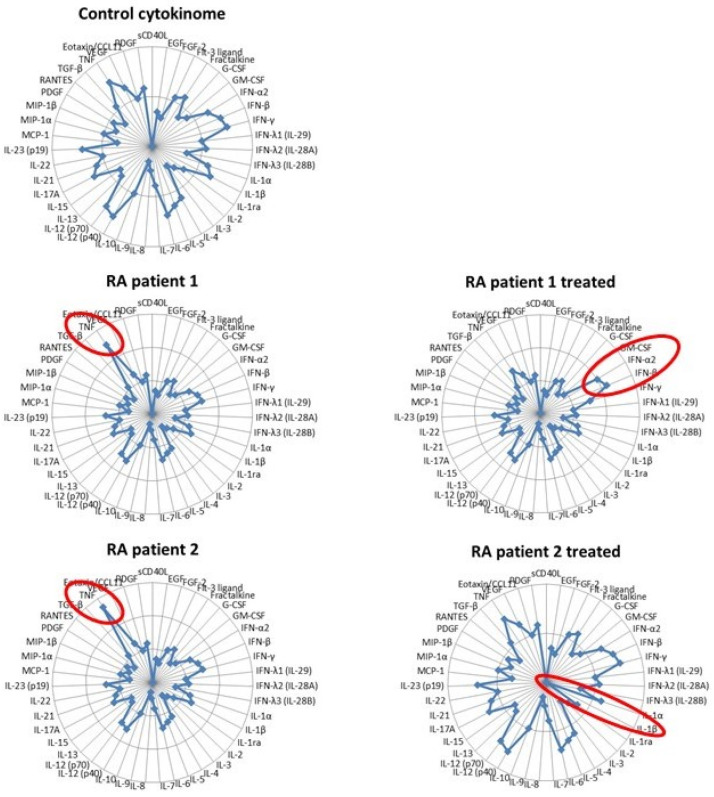
The multidimensional compass of inflammation. Radar plot showing the hypothetical expression levels of 43 cytokines/chemokines in the Control and two RA patients before and after anti-TNF therapy.

**Table 1 cells-10-01134-t001:** Type I IFNs- and IL-1β-mediated pathologies discussed in this review.

Disease	Type	Genetic Defect	Cytokine Profile	Treatment
STING-associated vasculopathy with onset in infancy (SAVI)	interferonopathy	STING gain-of-function	exessive type I IFN secretion	corticosteroids jakinhibs (clinical trials)
Systemic Lupus Erythematosus (SLE)	rheumatic autoimmune/autoinflammatory disease	multifactorial disease	IFN signature (overexpression of IFN-stimulated genes)	corticosteroids Immunosuppressants (e.g., methotrexate) biologics (e.g., antiB-cell mAb)
Familial Mediterranean Fever (FMF)	inflammasomopathy	mutations in MEFV (Mediterranean fever, also named PYRIN)	constitutive IL-1β secretion	colchicin biologics (IL-1β receptor antagonist, anti IL-1β mAb)
Alzheimer’s disease (AD)	Neurodegenerative disease	multifactorial disease	excessive IL-1β, IL-6 and TNF secretion	Cholinesterase inhibitors N-methyl D-aspartate (NMDA) antagonists anti amyloid-β mAb (clinical trials)
Gout	rheumatic autoinflammatory disease	multifactorial disease	excessive IL-1β secretion	colchicin biologics (IL-1β receptor antagonist, anti IL-1β mAb)
Rheumatoid Arthritis (RA)	rheumatic autoimmune/autoinflammatory disease	multifactorial disease	TNF overexpression IL-1β overexpression IFN signature (overexpression of IFN-stimulated genes)	corticosteroids Immunosuppressants (e.g., methotrexate) biologics (e.g., anti TNF mAb)
Multiple sclerosis (MS)	inflammatory, neurodegenerative disease	multifactorial disease	increased IFNγ, IL-12, IL-17 secretion/activation	IFN-β biologics (e.g., antiB-cell mAb)

## Data Availability

Not applicable.

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
