# Peer review of "Crosstalk between Interleukin-1β and Type I Interferons Signaling in Autoinflammatory Diseases"

_cells, 2021, doi:10.3390/cells10051134_

Round 1

Reviewer 1 Report

Philippe Georgel’s review titled Crosstalk between Interleukin-1β and type I Interferons signaling in autoinflammatory diseases.

The manuscript is very well written; clear, precise, and easy to understand.  I feel the overall manuscript has potential and has interesting for readers, I would be happy to suggest the editor accept the manuscript.

In Figure 1, The author mentioned Oversimplified representation of the potential interactions between plasmacytoid (pDC) or conventional (cDC) dendritic cell, macrophages/monocytes and neutrophils upon, for example, imiquimod (IMQ) stimulation acting via TLR7 in pDCs. Blue arrows denote cytokine expression…

How about the Eosinophils ?

Due to have reports related them as below.

Man Eosinophils Release IL-1β and Increase Expression of IL-17A in Activated CD4+ T Lymphocytes

IL-1β in eosinophil-mediated small intestinal homeostasis and IgA production

Anyway, Secretion of interleukin- (IL-1β) represents a fundamental innate immune response to microbial infection that…

I suggest the author talk about cGAS and IFNs in the Perspectives and Conclusions sections. The author also mentioned it in other part like this: Low STING-dependent type I IFNs expression in peripheral blood mononuclear cells (PBMC) isolated from MS patients [35] is in agreement with these observations.

References as below.

cGAS and Ifi204 Cooperate To Produce Type I IFNs in Response to Francisella Infection

Positive feedback regulation of type I IFN production by the IFN-inducible DNA sensor cGAS

Absence of cGAS-mediated type I IFN responses in HIV-1–infected T cells
The Vaccine Adjuvant Chitosan Promotes Cellular Immunity via DNA Sensor cGAS-STING-Dependent Induction of Type I Interferons

Author Response

I am thankfull to the referee for pointing to these issues.

1) Indeed, the role of eosinophils must be mentioned, although the isolation of these cells is highly challenging, which limits their use in cell culture systems. Nevertheless, following the referee's suffestion eosinophils have been added in the modified Figure (which is now Figure 2). A reference describing IL-1b production by these cells has also been included.

2) I also agree that the role of IL-1 beta-dependent cGAS activation and type I IFN expression is a major recent discovery. This has been included in the Perspectives section.

Reviewer 2 Report

The review by Georgel is focused in the role of type I IFNs and IL-1b in autoinflammatory diseases. First the author collects different inflammatory conditions with evidences of an “IFN signature”, or an “IL-1b signature”. Afterwards, the review focuses on the interplay between the two types of cytokines, and how IFNs can be targeted in IL-1b-related pathologies and vice versa. The review finalises with perspectives to dissect the role of both cytokines in different cell types. In the conclusions, the author points to the analysis of multiple cytokines in patients, as a strategy for stratification and possibly for the selection of a particular treatment. This review article is clearly and well written, and the different sections are in a logical order. However, the paper requires some minor changes prior to publication:

Comment 1. In the introduction, I miss a brief description of the key biological functions of IFNs and IL-1b in the context of anti-viral responses. This is relevant to understand if the pathogenic role of the cytokines in autoimmune disease is mediated by the same molecular mechanisms as during anti-viral response, just deregulated or excessive, or if different molecular mechanisms are triggered in the context of autoimmunity.

Comment 2. Taking into account that some of the therapeutic strategies discussed in the review aim to interfere with signalling cascades (i.e., Jakinibs), I miss a brief description or even a figure of IFNs and IL-1b signalling cascades (i.e., IFNs mediated by Jak/STAT, IL-1b by MyD88/IRAKs).

Comment 3. With the different diseases discussed in sections 2 and 3, the review will greatly benefit of a summary in a table format collecting all the information. For example, a table indicating the disease, if it is characterised by IL1b or type I IFN (ie. deregulated pathway or inflammatory signature), and the corresponding reference.

Comment 4. The section 4 is a key part of the review, as it describes different molecular mechanisms for the crosstalk between IFNs and IL-1b. I believe that a figure depicting the different mechanisms would increase the impact of the review.

Comment 5. Section 5 could benefit with the inclusion of a summary table collecting the different therapeutic strategies to target IFNs and IL-1b, the disease in which have been tested and the corresponding reference.

Comment 6. Minor typing errors.

Lines 52 and 107, missing references.

Line 101, written accent.

Line 203, dieresis.

Author Response

I am thankfull to the referee for his/her input in the manuscript and important advices and suggestions.

Comment 1. A short sentence with the appropriate references has been added in the Introduction to provide some information regarding the antiviral effects of type I IFNs and IL-1b. The reader might turn to these references to gain more information about the mode of action of these cytokines, which is not the purpose of the present review.

Comment 2. type I IFNs and IL-1b signalling has been extensively reviewed in dedicated reviews. However, the novel Figure 1, which summerizes the crosstalk between these cytokines, might partially answer the referee's demand.

Comment 3. A Table has been included to collect the informations on the diseases which are mentionned in the review.

Comment 4. As suggested by the referee, a figure has been added to schematically depict the type I IFNs/IL-1b crosstalk. 

Comment 5. Since the therapeutic options that are described in the review are only suggestions, such a Table would be misleading and eventually make the reader think that they are presently operational. I respectifully suggest not to include this type of Table in the manuscript.

line 52:  references have been added

line 107: a reference has been added

Line 101, written accent: correction is made

Line 203, dieresis: "organoïds" has been changed to "organoids"